# META-AWARENESS ENHANCES REASONING MODELS: SELF-ALIGNMENT REINFORCEMENT LEARNING

## ABSTRACT

Recent studies on reasoning models explore the meta-awareness of language models, the ability to know 'how to think' by itself. We argue that large reasoning models lack this meta-awareness property by proving severe misalignment between true rollouts and predicted meta information. We posit that aligning meta-prediction with true rollouts will lead to significant performance gains. To verify this hypothesis, we design a training pipeline that boosts Meta-Awareness via Self-Alignment (MASA), and prove that enhanced meta-awareness directly translates to improved accuracy. Unlike existing meta-cognitive reasoning models, our method does not require external training sources but leverages *self-generated signals to train meta-awareness*. Moreover, our method enables efficient training by i) filtering out zero-variance prompts that are either trivial or unsolvable and ii) cutting off lengthy rollouts when they are unlikely to lead to correct answers. The results are inspiring: our strategy yields significant improvements in both accuracy and training efficiency on in-domain tasks and shows strong generalization to out-of-domain benchmarks. More specifically, our method can speed up GRPO training by over $1.28\times$ to reach the same performance, and achieve a 19.3% gain in accuracy on AIME25, and a 6.2% average gain over six mathematics benchmarks. Training with meta-cognitive guidance enhances out-of-domain generalization, giving a 3.87 % boost on GPQA-Diamond and a 2.08 % overall accuracy gain across 13 benchmarks spanning logical, scientific, and coding domains.

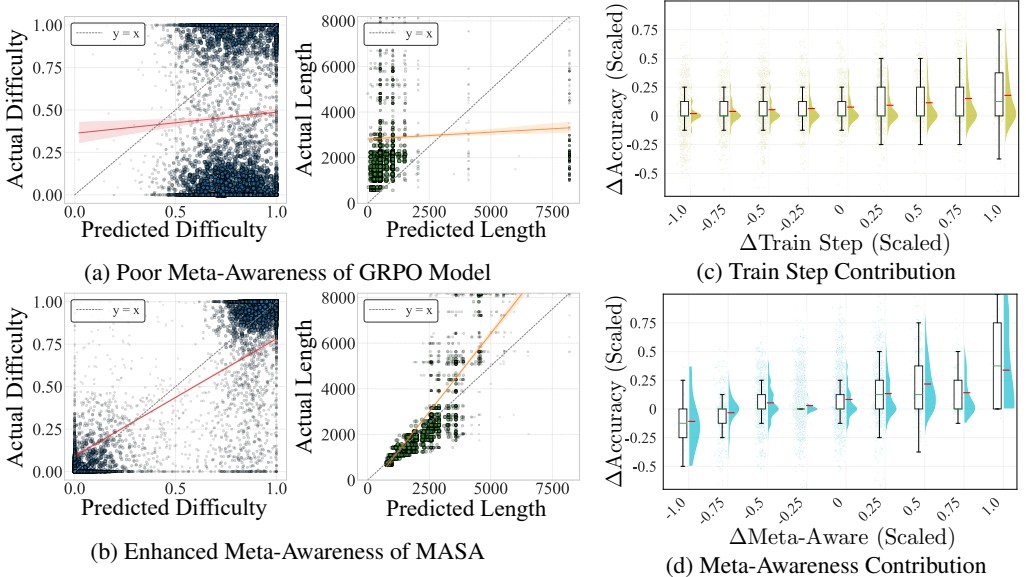

Figure 1: (a) Existing large reasoning models lack meta-awareness. (b) MASA significantly improves meta-awareness, as shown by the alignment between meta-predictions and the actual rollout statistics (difficulty and length). (c) Training step has limited impact on accuracy. (d) Meta-awareness directly translates to increased accuracy.

## 1 INTRODUCTION

Recent studies have confirmed that applying RL-based post-training to large language models (LLMs) (Brown et al., 2020; Yang et al., 2025a; Touvron et al., 2023) can significantly enhance their reasoning ability. In particular, methods such as GRPO (Shao et al., 2024), which efficiently train large reasoning models (LRMs) (Guo et al., 2025a; Chen et al., 2025b) without an explicit critic model, have recently attracted considerable attention. By directly incentivizing behaviors aligned with task-desirable outcomes, this training paradigm has gained prominence as an effective mechanism for attaining state-of-the-art performance on reasoning-intensive tasks such as mathematics and code generation.

Beyond the success of LRMs, the paradigm of meta-awareness, which is the ability to recognize it's own knowledge and ignorance, has drawn increasing attention from the research community (Sui et al., 2025; Ha et al., 2025; De Sabbata et al., 2024; Chen et al., 2025a; Liu et al., 2025b; Zhang et al., 2025a; Shen et al., 2025; Tu et al., 2025; Shi et al., 2025; Qu et al., 2025). However, existing approaches remain constrained by their reliance on external model, curated dataset and human-designed reasoning pipelines where meta-cognitive actions are only conditionally rewarded based on the success of the solution trajectory.

To this end, we propose a novel RL framework, Meta-Awareness via Self-Alignment (MASA), that strengthens the meta-awareness of reasoning models by rewarding the alignment within self-generated signals, eliminating the need for external sources. Our method further introduces parallel rollouts for meta-predictions and solution paths, separating them into distinct reward pipelines. We show that MASA improves reasoning performance by leveraging meta-awareness of solution length, problem difficulty, and underlying mathematical concepts, outperforming even the gains achieved by simply increasing training steps (Figure 1c, Figure 1d).

To strengthen the alignment between actual rollout statistics and meta-predictions, we introduce supervised fine-tuning on dynamically collected expert meta-trajectories, following a DAgger-style imitation learning approach (Ross et al., 2011). The improved meta-predictions make training more efficient through *predictive gating*, which identifies and filters out zero-variance prompts that are either trivial or unsolvable, and *early cutoff*, which terminates long rollouts that are predicted to be incorrect. In addition, the meta-predictions enrich prompts with auxiliary hints that facilitate reasoning.

Building on this foundation, we evaluate the effectiveness of our approach by combining with GRPO and DAPO (Yu et al., 2025; Shao et al., 2024), showing that our method is not dependent on specific policy gradient algorithm. Remarkably, MASA achieves substantial improvements in in-domain mathematical benchmarks showing average accuracy gains of 6.2%. Furthermore, boosting meta-awareness also enhances generalization, as evidenced by improvements across logical, coding, and scientific reasoning benchmarks. These results demonstrate that equipping reasoning models with meta-awareness not only strengthens in-domain performance but also broadens general reasoning capabilities. Finally, predictive gating and early cutoff deliver significant efficiency gains, attaining baseline performance 1.28 times faster than the GRPO training.

The contributions of this paper can be summarized as follows:

- We demonstrate that enhancing meta-awareness directly translates into measurable performance gains on complex reasoning tasks.
- We demonstrate that incentivizing meta-awareness improves both in-domain and out-of-domain generalization across logical, scientific, and coding benchmarks.
- We show the efficacy of meta-prediction based post-training via predictive gating and early cutoff, speeding up the time to reach baseline performance by $1.28\times$.

## 2 RELATED WORKS

**Meta-Cognitive Learning** Meta-cognition is viewed as a prerequisite for self-improving LLMs (Liu & van der Schaar, 2025). Existing methods rely on extrinsic mechanisms with fixed action loops, limiting adaptability. Self-improving agents that plan, regulate, and reflect (Dong et al., 2025; Didolkar et al., 2025) or refine prompts via past reasoning (Qiu et al., 2025; Liu et al., 2025b)

(a) Self-Alignment Reward (section 3.2)   (b) Meta-Aware Gating & Hinting & Cutoff (section 3.3)

Figure 2: **Overall Framework of MASA** (a) Parallel rollout of meta prediction path and solution path. Meta predictions are rewarded by self-alignment from statistics collected from solution roll-outs. (b) Meta-based predictive gating, early cutoff and notion hinting from meta-predictions.

entangle control with reasoning, often causing interference. In contrast, our approach disentangles the meta and solution path separately for stable training on meta-awareness.

Other works require curated datasets (Ha et al., 2025), or delegate control to external verifiers (Ma et al., 2025; He et al., 2025) or multi-agent systems (Wan et al., 2025; Yang & Thomason, 2025; Bilal et al., 2025; Khandelwal et al., 2025), reducing scalability of meta-cognitive training. Training-free heuristics such as confidence-based stopping (Yang et al., 2025b; Qiao et al., 2025; Lu et al., 2025) or correctness checks (Ma et al., 2025) offer efficiency but lack genuine language-level meta-cognition. In contrast, our approach do not rely on human-curated reasoning pipelines, external verifiers/PRMs, or specialized datasets targeting meta-cognitive ability, but rather leverage the *self-generated signals to encourage alignment* between the meta-prediction and primary thinking process.

**Self-Control for Efficient Training** Another direction that leverages meta-cognition is to regulate reasoning efficiency by allocating budgets via difficulty assessment (Chen et al., 2025a; Tu et al., 2025; Shi et al., 2025; Qu et al., 2025; Huang et al., 2025; Ji et al., 2025; Di & JoyJiaoW, 2025; Han et al., 2024b; Fang et al., 2025; Yang et al., 2025c; Zhang et al., 2025b; Wang et al., 2025; Zhang et al., 2025a; Shen et al., 2025), constraining output length with penalties or fixed limits (Aggarwal & Welleck, 2025; Li et al., 2025; Xiang et al., 2025; Zhang & Zuo, 2025), and adaptively stopping, continuing, or reflecting for compact reasoning (Ha et al., 2025; Zhang et al., 2025c; Dai et al., 2025). While these methods improve inference-time efficiency, they focus on making reasoning shorter or faster at inference time, often at the expense of reasoning performance drop. In contrast, we target *post-training efficiency*, achieving both efficiency and improved performance during model training rather than the inference.

## 3 MASA: META-AWARENESS VIA SELF-ALIGNMENT AND MASA-*efficient*

We first provide background on group relative policy optimization (GRPO) variants (Section 3.1). Then we show our method: (i) MASA, which endows the LLM with the capability to perform accurate meta-predictions (Section 3.2); and (ii) MASA-*efficient*, an efficiency-enhanced version that accelerates MASA through predictive gating, early cutoff, and prompt hinting (Section 3.3).

### 3.1 PRELIMINARIES

We present an overview on GRPO, which is a popular RL algorithm for post-training reasoning models. Given a task $q$ drawn from the distribution $\mathcal{Q}$, the policy model $\pi_{\theta_{old}}$ produces a group of $G$ responses, which are referred to as rollouts, $\{o_1, \cdots, o_G\}$. Each response is assigned with a reward $\{r_1, \cdots, r_G\}$ based on the match between the ground truth answer and the extracted answer from the response. This is formalized as

$$\mathcal{L}_{RL}(\theta) = \mathbb{E}_{q \sim \mathcal{Q}, \, \{o_i\}_{i=1}^{G} \sim \pi_{\theta_{old}(\cdot|q)}}$$

$$\left[ \frac{1}{G} \sum_{i}^{G} \frac{1}{|o_i|} \sum_{t}^{|o_i|} \left\{ \min \left[ \Gamma_{i,t}(\theta) \hat{A}_{i,t} \,, \, \text{clip}_{1-\epsilon}^{1+\epsilon}(\Gamma_{i,t}(\theta)) \hat{A}_{i,t} \right] - \beta D_{KL}(\pi_\theta \| \pi_{ref}) \right\} \right],$$

where the importance sampling ratio between the current policy $\pi_\theta$ and the old policy $\pi_{\theta_{old}}$ is defined as $\Gamma_{i,t}(\theta) = \pi_\theta(o_{i,t} \mid q, o_{i,<t})/\pi_{\theta_{old}}(o_{i,t} \mid q, o_{i,<t})$, and $\text{clip}(\cdot)$ restricts the importance sampling ratio between $[1-\epsilon, 1+\epsilon]$. Advantage calculation is formulated as $\hat{A}_{i,t} = \frac{r_i - \text{mean}(\{r_i\}_{i=1}^G)}{\text{std}(\{r_i\}_{i=1}^G)}$. Following the practice of recent RL algorithms proposed in recent GRPO variants (Liu et al., 2025a; Zhang & Zuo, 2025; Zheng et al., 2025; Yu et al., 2025), we set $\beta = 0$ to ignore the KL divergence term.

## 3.2 MASA: META-AWARENESS VIA SELF-ALIGNMENT

The policy model $\pi_\theta$ is prompted with the task $q$ with two variants of instruction templates, meta-prediction template and solution template, creating $q_{\text{meta}}$ and $q_{\text{sol}}$[1]. The policy model outputs meta-prediction rollouts $\{o_i^{\text{meta}}\}_{i=1}^M$ given $q_{\text{meta}}$ and solution rollouts $\{o_i^{\text{sol}}\}_{i=1}^G$ given $q_{\text{sol}}$ in parallel. The solution rollouts are equivalent to the rollouts in regular GRPO algorithm explained in Section 3.1, while meta rollouts are structured responses that consist of predicted length, predicted difficulty, and the list of mathematical notions.

The rollout and reward assignment for solution rollouts and meta-predictions are separated as described in Figure 2(a). For solution, the reward is assessed by the agreement between model's solution and the ground truth solution, which we denote as $\{r_i^{\text{sol}}\}_{i=1}^G$. For meta-prediction rollouts, we rely on three rewarding criteria: self-alignment of length, pass-rate, and math notions, averaged into $r_{\text{meta}} = (r_{\text{length}} + r_{\text{difficulty}} + r_{\text{notion}})/3$.

**Length Reward.** The length alignment reward assigns 1 if the prediction belongs in the range of rollout lengths of correct solution paths. More formally, we define the length reward as

$$r_{\text{length}} = \mathbb{1}\big[\min(\mathbf{l}_{\text{correct}}) \le l_{\text{pred}} \le \max(\mathbf{l}_{\text{correct}})\big], \tag{1}$$

where $\mathbf{l}_{\text{correct}}$ is a list of correct response lengths from solution rollouts $\{o^{\text{sol}}\}$ and $l_{\text{pred}}$ is the predicted length from meta rollout. In cases where correct responses do not exist for the task $q$ ($|\mathbf{l}_{\text{correct}}| = 0$), then the reward assigned becomes 0.

**Difficulty Reward.** The difficulty alignment reward is computed as exponentially decaying reward by the factor of difference between the predicted pass-rate $d_{\text{pred}}$ and the true pass-rate $d_{\text{sol}}$ as

$$r_{\text{difficulty}} = b^{|d_{\text{pred}} - d_{\text{sol}}|} \tag{2}$$

where $b < 1$. We choose an exponentially decaying reward to ensure that the reward becomes 1 if $|d_{\text{pred}} - d_{\text{sol}}| = 0$ and rapidly approach to 0 as the difficulty difference becomes larger.

**Notion Reward.** The notion reward is defined for the list of notions, $\mathbf{n}_{\text{pred}} = [n_1, \cdots, n_p]$, which are mathematical concepts that are predicted to be used in solution rollout that yields correct answer. We count the ratio of notions that appear more frequently in correct solution rollouts than in incorrect ones. Formally we define notion reward as

$$r_{\text{notion}} = \frac{1}{|\mathbf{n}_{\text{pred}}|} \sum_{n \in \mathbf{n}_{\text{pred}}} \mathbb{1}\big[f_{\text{count}}(n, 1) - f_{\text{count}}(n, 0) > 0\big], \tag{3}$$

where $f_{\text{count}}$ is a function that counts the number of notion appearance in correct or incorrect solution rollouts. The counting function is defined as follows,

$$f_{\text{count}}(n, t) = \big|\{i \in \{1, \ldots, G\} : n \in o_i^{\text{sol}}, r_i^{\text{sol}} = t\}\big|, \qquad t \in \{0, 1\}, \tag{4}$$

to reward a notion $n$ that is more frequently included in correct solutions ($t = 1$) than in incorrect ones ($t = 0$). In detail, the notions included in the problem itself is excluded in the counting process to avoid reward hacking and the predicted notions are lemmatized to properly find inclusion in the solution rollouts via exact matching.

---

[1]The average token length of meta-predictions are 36% of average solution rollout length. The meta-prediction template is deferred to Appendix A.

---

**Algorithm 1** MASA-*efficient*: Efficient Meta-Aware Training with SFT on Expert Trajectories.

---

**Require:** Task distribution $\mathcal{Q}$, expert dataset buffer $\mathcal{D}_{\text{expert}}$, initial policy parameters $\theta$, efficient start step $k$
**Ensure:** Optimized policy parameters $\theta$
    $\theta_{\text{old}} \leftarrow \theta$
    **for** step $= 1, \ldots, N$ **do**
        Sample task prompt $q \sim \mathcal{Q}$                              ▷ or a minibatch
        Sample meta-trajectories $\{\boldsymbol{o}_i^{\text{meta}}\}_{i=1}^M \sim \pi_{\theta_{\text{old}}}(\cdot \mid q_{\text{meta}})$
        **if** step $> k$ **then**
            Efficient sampling with predictive gating, early cutoff, and notion hinting
        **else**
            Sample reasoning trajectories $\{\boldsymbol{o}_i^{\text{sol}}\}_{i=1}^G \sim \pi_{\theta_{\text{old}}}(\cdot \mid q_{\text{sol}})$
        **end if**
        $\theta \leftarrow \theta - \alpha \nabla_\theta \mathcal{L}_{\text{RL}}(\theta)$
        Extract expert trajectory $\{\mathbf{o}_{\text{expert}}\}$ from $\{\boldsymbol{o}_i^{\text{sol}}\}_{i=1}^G$ and $\{\boldsymbol{o}_i^{\text{meta}}\}_{i=1}^M$
        $\mathcal{D}_{\text{expert}} \leftarrow \mathcal{D}_{\text{expert}} \cup \{\mathbf{o}_{\text{expert}}\}$
        **if** $|\mathcal{D}_{\text{expert}}| \geq N_{\text{expert}}$ **then**
            $\theta \leftarrow \theta - \beta \nabla_\theta \mathcal{L}_{\text{BC}}(\theta, \mathcal{D}_{\text{expert}})$                  ▷ Equation (5)
            $\mathcal{D}_{\text{expert}} \leftarrow \emptyset$
        **end if**
        $\theta_{\text{old}} \leftarrow \theta$
    **end for**

---

## 3.3 MASA-*efficient*: META-BASED ACTIVE CONTROL FOR EFFICIENT POST-TRAINING

MASA-*efficient* is a variant of MASA that can further boost training efficiency by leveraging the length and difficulty predictions from meta-predictions. From the observation that early step meta-predictions are unstable, we encourage the behavior cloning of the policy model on the ideal meta-prediction trajectories that are gathered throughout each RL step, inspired by behavior cloning (BC) (Mendonca et al., 2019; Silver et al., 2017; Schick et al., 2023). We denote these ideal meta-predictions as expert dataset, $\mathcal{D}_{\text{expert}}$, which are meta-predictions that scored high notion score and the predictions on pass-rate and length are substituted by the true statistics gathered from the solution rollouts. Once the expert dataset size reaches $N_{\text{expert}}$, we minimize cross-entropy loss on $\mathcal{D}_{\text{expert}}$ on the current policy model as

$$\min_\theta \mathcal{L}_{\text{BC}}\left(\theta - \alpha \nabla_\theta \mathcal{L}_{\text{RL}}(\theta), \mathcal{D}_{\text{expert}}\right), \tag{5}$$

where $\alpha$ is the learning rate for RL training. Formally, the behavior cloning loss is defined as $\mathcal{L}_{\text{BC}}(\theta, \mathcal{D}_{\text{expert}}) = \mathbb{E}_{\boldsymbol{o} \sim \mathcal{D}_{\text{expert}}} \left[ -\sum_{t=1}^{|\boldsymbol{o}|} \log \pi_\theta\left(o_t \mid \boldsymbol{o}_{<t}\right) \right]$.

Note that we gather samples from the current policy model prediction and accumulate up to a batch size of $N_{\text{expert}}$, as outdated trajectories do not reflect the current policy model behavior, and the outdated expert meta trajectories are evicted from $\mathcal{D}_{\text{expert}}$ following DAgger (Ross et al., 2011). We prove the boundedness of cost-to-go of this algorithm to ensure that the policy model improves stably in Appendix C.

**Non-Parallel Efficient Training with Gating and Cutoff** As MASA-*efficient* is the efficient variant of MASA. To encourage meta-awareness before accelerating the training phase, we first perform self-alignment based policy updates for the early $k$ steps following MASA pipeline[2], until the policy model shows stable meta-prediction alignment with the true solution rollouts. From this point, we alter into non-parallel pipeline that executes meta-predictions first, for **predictive gating**, followed by solution rollouts, applying **early length cutoff**. We also utilize the predicted notions to provide additional hint for the model in solving the questions as illustrated in Figure 2(b).

Predictive gating filters out zero-variance tasks, that exceeds or under-reaches the model's current capacity. Unlike DAPO that performs pruning after doing lengthy and inefficient solution rollouts, our method saves computation by using short meta-predictions as a gate on whether to rollout the lengthy solution beforehand[3]. In detail, the predictive gating is activated only if the standard deviation over $M$ predicted pass-rates is below $0.1$ to ensure confident meta-prediction. The length

---

[2]The selection of training step $k$ is explained in Figure 3.
[3]The length difference between the meta and solution rollouts is analyzed in Table 3.

Table 1: **Performance of GRPO and MASA across In-domain Math benchmarks.**

| Benchmark | GRPO | | GRPO w/ MASA | |
|---|---|---|---|---|
| | Pass@1 | Pass@32 | Pass@1 | Pass@32 |
| **Qwen3-14B Base Model** | | | | |
| AIME'24 | 38.54 | 70.00 | **40.10** (+ 4.04%) | **73.33** (+ 4.76%) |
| AIME'25 | 27.91 | 56.67 | **29.90** (+ 7.13%) | 56.67 ( - ) |
| AMC'23 | 81.56 | 97.50 | **84.61** (+ 3.74%) | 97.50 ( - ) |
| MATH500 | **88.61** | 97.60 | 88.54 (- 0.08%) | **97.80** (+ 0.20%) |
| Minerva | 44.84 | 71.32 | **45.37** (+ 1.18%) | **74.63** (+ 4.64%) |
| Olympiad | 58.65 | **77.74** | **59.94** (+ 2.20%) | 77.15 (- 0.76%) |
| **Average** | 56.69 | 78.47 | **58.08** (+ 2.45%) | **79.51** (+ 1.33%) |
| **Qwen3-8B Base Model** | | | | |
| AIME'24 | 28.54 | 66.67 | **33.75** (+ 18.26%) | **70.00** (+ 5.00%) |
| AIME'25 | 22.18 | 46.67 | **26.46** (+ 19.30%) | **50.00** (+ 7.14%) |
| AMC'23 | 73.67 | 97.50 | **76.88** (+ 4.36%) | **100.00** (+ 2.56%) |
| MATH500 | 85.75 | 96.80 | **87.36** (+ 1.88%) | **96.80** ( - ) |
| Minerva | 42.46 | 69.85 | **45.35** (+ 6.81%) | **72.06** (+ 3.16%) |
| Olympiad | 53.61 | 76.11 | **55.41** (+ 3.36%) | **78.48** (+ 3.11%) |
| **Average** | 51.04 | 75.60 | **54.20** (+ 6.20%) | **77.89** (+ 3.03%) |

prediction is used as a early cutoff threshold to stop the rollout that exceeds more than $2\times$ of the predicted length, as such lengths are highly likely to lead to incorrect rollout due to notion reward design. The precision and F1 score of predictive gating and early length cutoff in predicting the true zero-variance and incorrect rollouts are analyzed in Figure 3.

## 4 EXPERIMENTS

**Implementation Details.** We use VeRL with the DeepScalerR (Luo et al., 2025) dataset, batch size 128, learning rate 1e-6, 10% weight decay, maximum response length 8K, and GRPO without KL. Training runs for one epoch (314 steps) using AdamW (Loshchilov & Hutter) with 20 warm-up steps, gradient clipping 1.0, and clipping range $[\epsilon_{\text{low}} = 0.2, \epsilon_{\text{high}} = 0.28]$. The rollouts use temperature 1.0 and top-p value of 1.0. Both actual ($G$) and meta-prediction ($M$) rollouts are 16. Expert SFT uses 5 gradient updates per outer RL loop. The difficulty-reward base is $b = 0.01$, and gating/cutoff begins at $k = 120$ and the batch size for expert dataset is also set as 128.

**Evaluation Configuration.** We use the provided math scoring function in VeRL to measure the accuracy of the predicted answer and ground truth answer sampling 32 responses, 16k maximum response length and temperature 0.6.

**Baselines.** The baseline of our method is GRPO and DAPO. Throughout the experiment section, **MASA** refers to the model that is trained with our Meta-Awareness via Self-Alignment. **MASA-efficient** indicates the version of a model that includes the gating & cutoff applied from MASA at step 120.

### 4.1 OBSERVATIONS

We analyze the performance of MASA through validation on mathematical benchmarks and generalized reasoning benchmarks.

**MASA Excels in In-Domain Mathematical Benchmarks**  MASA excels the baseline in six math benchmarks, AIME24, AIME25, AMC23, MATH500 (Hendrycks et al.), Minerva, and Olympiad-Bench (He et al., 2024) (Table 1). Across all mathematical datasets, our method MASA shows great improvement over the baseline GRPO performance, showing an average of 6.2% improvement in Qwen3-8B model, and an average of 2.45% in 14B model.

Table 2: **Performance of GRPO and MASA in Out-of-Domain benchmarks.** Results are reported as pass@1 score.

| Logical Reasoning | | | Scientific Reasoning | | | Coding | | |
|---|---|---|---|---|---|---|---|---|
| **Benchmark** | **GRPO** | **w/ MASA** | **Benchmark** | **GRPO** | **w/ MASA** | **Benchmark** | **GRPO** | **w/ MASA** |
| ProntoQA | 90.56 | **93.74** | GPQA Diamond | 51.72 | **53.72** | EvalPlus | 77.32 | **77.66** |
| ProofWriter | 72.27 | **73.23** | R-Bench | 60.69 | **61.68** | CRUX-O | 72.72 | **73.39** |
| FOLIO | 69.16 | **69.24** | ARC-Challenge | 93.10 | **93.13** | MBPP | 71.84 | **72.97** |
| Logi. Deduct | 80.81 | **81.03** | SciBench | 28.33 | **29.64** | LiveCodeBench | 31.49 | **31.61** |
| AR-LSAT | 37.00 | **38.00** | | | | | | |
| **Avg.** | 69.96 | **71.05** | **Avg.** | 58.46 | **59.54** | **Avg.** | 63.34 | **63.91** |

**MASA Generalizes to Out-of-Domain Reasoning Benchmarks** The meta-awareness also benefits generalization ability of the reasoning model in out-of-domain logical & scientific & coding benchmarks as shown in Table 2. For logical reasoning domain, we follow the setup of (Pan et al., 2023) and test on ProntoQA (Saparov & He), ProofWriter (Tafjord et al., 2021), FOLIO (Han et al., 2024a), LogicalDeduction (Srivastava et al.), and AR-LSAT (Zhong et al., 2022). For scientific reasoning, we use GPQA Diamond (Rein et al., 2024), R-Bench (Guo et al., 2025b), ARC-Challenge (Clark et al., 2018), and SciBench (Wang et al., 2024). For coding, we evaluate on EvalPlus (Liu et al., 2023), CRUX-O (Gu et al., 2024), MBPP (Austin et al., 2021), and LiveCodeBench (Jain et al., 2025). Although MASA is not explicitly trained for generalization, strengthening meta-awareness consistently enhances out-of-domain performance.

## 4.2 ANALYSIS ON COMPONENT

**Implicit Meta-Awareness Reward Explicitly Changes the Model Output.** How does the parallel rollout of meta-predictions influence the solution rollouts? We classify notions that appear more often in correct responses as positive notions and those that appear more often in incorrect responses as negative notions. After reward-based gradient updates, positive notions should become more common in correct solution rollouts, whereas negative notions should be suppressed. As shown in Figure 3a, positive notions from earlier steps consistently increase in correct rollouts (notion score $> 0$ indicates higher frequency in correct compared to incorrect), whereas negative notions are reduced in correct rollouts but amplified in incorrect ones (notion score $< 0$).

**Expert Trajectories Increases Meta-Awareness in Early Train Steps.** Predictive gating aims to identify zero-variance prompts before rollout, while early cutoff predicts rollouts that will yield incorrect answer despite excessive token length. Adding expert trajectory supervised finetuning to MASA improves the precision of both mechanisms, as shown in Figure 3b and Figure 3c. Without expert SFT, MASA (green) shows unstable precision that drops sharply around step 80 and score F1 score of 0.411 and 0.732 in predictive gating and early cutoff, respectively. In contrast, with expert trajectories stabilizes the improvement, yielding final F1 score of 0.485 and 0.836 at training step 120. Based on this analysis, we begin to apply gating and cutoff only after step 120, once the predictions are stable in terms of both precision and F1 score.

**MASA-*efficient* reaches higher performance faster with faster train time.** Table 3 shows the effectiveness of MASA-*efficient* in reducing the train time compared to MASA. The train time drastically reduces by $34.5\%$, while closely retaining the performance of MASA in intermediate level of math reasoning tasks such as AMC23 and MATH500. On the other hand, MASA-*efficient* shows at most 3.9% of performance drop in AIME, which consists of Olympiad level math problems, proving the need for less efficient but stronger MASA for complex reasoning tasks.

We observe efficiency in terms of number of training tasks, number of generated tokens, and train time in Figure 4. MASA-*efficient* reaches the performance of the baseline model GRPO with notably smaller number of tasks, total generated tokens, and train time. As shown in the figure, the accuracy consistently outperforms the baseline under same budget condition, proving that MASA-*efficient* is highly effective in reducing the train time and compute resource. It is important to note that though our method MASA requires doubled rollouts for solution and meta-prediction paths, the average

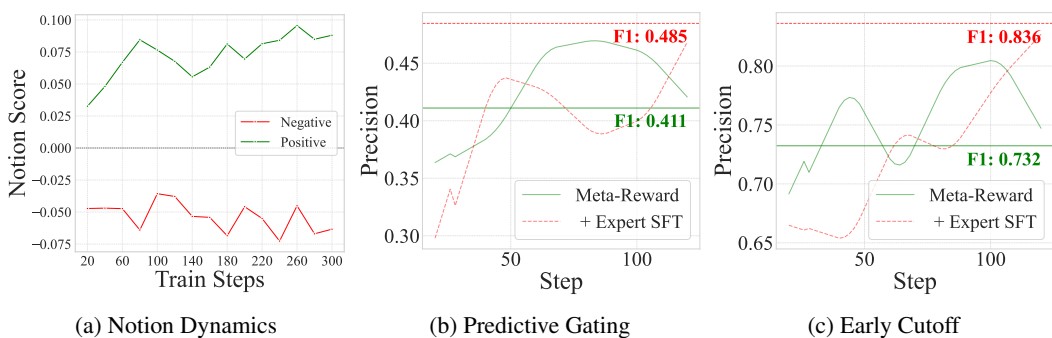

(a) Notion Dynamics          (b) Predictive Gating          (c) Early Cutoff

Figure 3: (a) Notion score of positive / negative notions from earlier train step. (b) Precision Score of Predictive Gating on true zero variance prompts. (c) Precision Score of Early Cutoff on true incorrect roll-outs. Precisions are smoothed by a moving average over 5 steps.

Table 3: Analysis on MASA-*efficient* performance and average token length of two trajectories with MASA.

|  | MASA | MASA-*efficient* | Perf. Gap |
|---|---|---|---|
| AIME'25 | 33.75 | 32.71 | -3.1% |
| AIME'24 | 26.46 | 25.42 | -3.9% |
| AMC23 | 76.88 | 76.88 | - |
| MATH500 | 87.36 | 87.68 | +0.4% |
| Avg | 56.11 | 55.67 | -0.7% |
| Train Time (hrs) | 52.50 | 34.93 | -34.5% |

(a) Performance and efficiency comparison.

|  | Solution Path | Meta-Pred Path |
|---|---|---|
| Token Length | 6251 | 2293 |

(b) Average token length.

meta length of 2293 is 2.73 times smaller than the average solution path of average length 6251. By adding predictive gating and length cutoff, the total train time becomes much shorter since the gating happens before the lengthy solution path.

Figure 5 shows the average proportion of prompts filtered by gating. On average, about 37% of prompts are removed before the model begins its full solution rollout, with the gating rate typically staying between 20–40%. Early in the process, up to 80% of prompts remain after gating, but this quickly drops to a stable, lower level. Although the baseline and MASA process the same number of tasks up to step 120, only 56% of tasks remain after filtering when using MASA-*efficient* after step 120 until step 314, compared to GRPO. Finally, note that we cannot measure the exact amount of rollout length saved by early cutoffs, since truncated rollouts do not reach an EOS token and thus their full length is unknown.

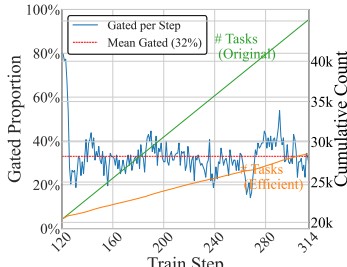

Figure 5: **Analysis on Gating.**

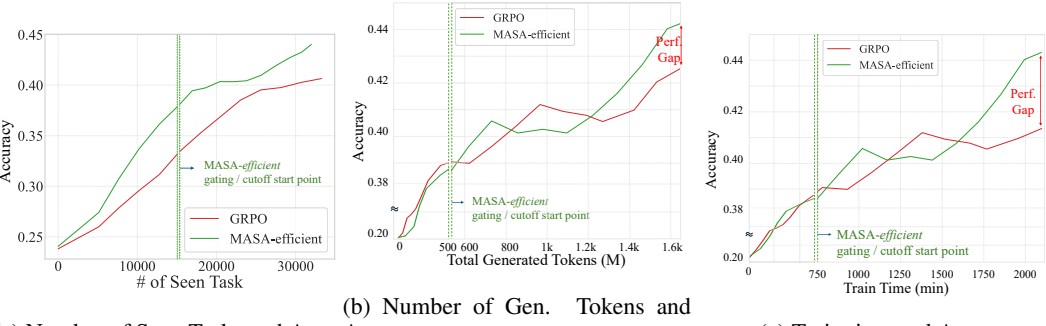

(a) Number of Seen Tasks and Acc.          (b) Number of Gen.   Tokens and Acc.          (c) Train time and Acc.

Figure 4: Comparison of MASA-*efficient* and GRPO on same train budgets: number of seen train tasks, total generation tokens, and train time. Accuracy is calculated as the average of AIME'24, AIME'25, and AMC'23. All accuracy curves are smoothed with a 3-window moving average.

## 4.3 ABLATION STUDIES

**Ablation on RL Algorithm**    We test the applicability of our method MASA on DAPO in Table 4, which is a variant of GRPO that introduces several technical changes in the optimization process. DAPO uses dynamic sampling to filter out tasks that yield zero-variance prompts to stabilize the gradient update and assigns penalty on overlong responses. We observed that applying the overlong penalty adversely affected accuracy under the 8k maximum response length setting. Accordingly, we adopted DAPO without the overlong penalty as the baseline. For DAPO, we conducted training for one epoch, consistent with the GRPO setup, and report the performance of the final model. Combined with DAPO, our method MASA outperforms all six mathematical benchmarks, reaching 18.61% of gain on Pass@1 in AIME'24.

Table 4: Performance comparison of MASA with DAPO, trained with Qwen3-8B base model.

| | **DAPO** | | **DAPO + MASA** | |
|---|---|---|---|---|
| **Benchmark** | Pass@1 | Pass@32 | Pass@1 | Pass@32 |
| AIME'24 | 23.54 | 63.33 | **27.92** (+ 18.61%) | **70.00** (+ 10.53%) |
| AIME'25 | 18.75 | 46.67 | **20.63** (+ 10.03%) | **60.00** (+ 28.56%) |
| AMC'23 | 67.11 | 97.50 | **69.22** (+ 3.14%) | 97.50(-) |
| MATH500 | 81.67 | **96.80** | **82.99** (+ 1.62%) | 96.20 (- 0.62%) |
| Minerva | 35.53 | 68.01 | **39.66** (+ 11.62%) | **71.69** (+ 5.41%) |
| Olympiad | 49.30 | 75.07 | **50.93** (+ 3.31%) | **76.71** (+ 2.18%) |
| **Average** | 45.98 | 74.56 | **48.56** (+ 5.61%) | **78.68** (+ 5.53%) |

**Meta-Component Contribution.**   Here we analyze which component among the three meta-predictions contribute the most to the performance increase. The contribution of length, difficulty, and notion prediction for meta-awareness is shown in Figure 6. In specific, we analyze the Shapley $R^2$ share[4] of each feature - the three meta components (notion-aware, difficulty-aware, length-aware), and the train step - on the contribution to the increase in model performance. The results show that notion-awareness is by far the most dominant factor, explaining over two-thirds of the variance in performance increase. Difficulty-awareness and length-awareness plays smaller role while the effect of training step is almost negligible.

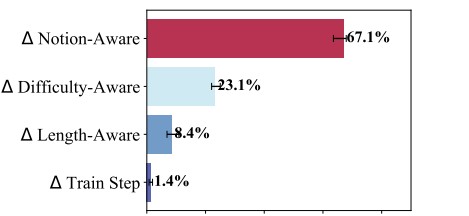

Figure 6: **Analysis on Meta-Components.**

## 5 CONCLUSION

We present MASA, a meta-aware reinforcement learning framework that fosters meta-cognitive ability by self-alignment. By incorporating expert meta-thinking trajectories into training, our method enables stable and efficient optimization by integrating predictive gating and early cutoff. Empirically, MASA accelerates RL-based post-training while improving both in-domain and out-of-domain performance, demonstrating notable gains in accuracy and generalization. These results highlight the promise of meta prediction as a principled avenue for enhancing reasoning models.

## LIMITATION

While our approach to meta prediction can, in principle, be extended to a broader range of meta-thinking strategies, in this work we focus on length, difficulty, and notion. The gating and cutoff hyper-parameters are set offline based on the analysis, but it would be beneficial to search hyper-parameters online during train time.

---

[4]Calculated by LMG (Lindeman–Merenda–Gold) method.

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

## A    DEFAULT META-PREDICTION PROMPT FOR MASA

> **Prompt**
>
> **[System]:**
> You are a helpful assistant.
> **[User]:**
> Think step-by-step between <meta> and </meta>, ensuring comprehensive and detailed reasoning especially for determining the pass_rate and solution_length values. For each component (math_notion, pass_rate, solution_length), provide a comprehensive illustration or example during your reasoning in the <meta> section to clarify how each value is decided. When determining math_notion, ensure that the notions listed do not directly include the notions already written in the problem statement. After </meta>, return a JSON object with three keys:
> - math_notion (list[str])
> - pass_rate (integer from 0 to 8)
> - solution_length (integer from 128 to {max_response_length})
>
> Problem: {problem}

## B    EFFECT OF NOTION FEED-IN FOR MASA INFERENCE

During training of MASA-*efficient*, we provided hints to the model through meta-predictions, whereas evaluation was conducted using only the actual rollout. Nevertheless, it is also possible to incorporate the notions predicted by the meta-prediction rollout into the prompt during inference, mirroring the training procedure. To examine the impact of such notion feed-in on performance, we performed the following experiment.

We extended the training pipeline of MASA with Expert SFT by appending the notions predicted by meta-prediction to the original prompt as additional context. We then compared the final model's performance with and without notion feed-in at inference time. The results are presented in table 5.

As shown in the table, although the improvements are modest, incorporating notion feed-in consistently yields slightly higher Pass@1 scores on most benchmarks. This finding suggests that the predicted notions can serve as useful cues for problem solving and may enable further performance gains when leveraged during inference.

Table 5: **Performance of GRPO and MASA on Qwen3-8B across In-domain Math benchmarks.** All metrics are **Pass@1**. "NF" denotes Notion-FeedIn.

| Benchmark | GRPO Pass@1 | GRPO w/ MASA Pass@1 | MASA + Expert (No NF) Pass@1 | MASA + Expert (NF) Pass@1 |
|---|---|---|---|---|
| **Qwen3-8B Base Model** | | | | |
| AIME'24 | 28.54 | 33.75 | 32.92 | **33.85** |
| AIME'25 | 22.18 | **26.46** | 26.04 | **26.46** |
| AMC'23 | 73.67 | 76.88 | 76.64 | **78.98** |
| MATH500 | 85.75 | 87.36 | 87.65 | **87.72** |
| Minerva | 42.46 | 45.35 | **46.43** | 45.86 |
| Olympiad | 53.61 | 55.41 | 55.07 | **55.59** |
| **Average** | 51.04 | 54.20 | 54.13 | **54.74** |

## C  BOUNDED RETURN-TO-GO OF EXPERT SFT

Let $\mathcal{Q}$ be a distribution over tasks $q$. Each task is a finite-horizon process of length $H$ that produces an output sequence $O = (o_1, \ldots, o_H)$ over a finite vocabularies $\mathcal{V}$. Write $o_{<t} := (o_1, \ldots, o_{t-1})$ for the history (prefix) at step $t$. GRPO objective over tasks is defined as

$$\mathcal{L}_{\mathrm{RL}}(\theta) = \mathbb{E}_{q \sim \mathcal{Q}, \, \{o_i\}_{i=1}^G \sim \pi_{\theta_{\mathrm{old}}}(\cdot \mid q)}$$

$$\left[ \frac{1}{H} \sum_{t=1}^H \underbrace{\left\{ \min \left[ \Gamma_{i,t}(\theta) \hat{A}_{i,t} , \mathrm{clip}_{1-\epsilon}^{1+\epsilon}(\Gamma_{i,t}(\theta)) \hat{A}_{i,t} \right] \right\}}_{r_t(\pi_\theta)} \right], \quad (6)$$

where $\Gamma_t(\theta) := \dfrac{\pi_\theta(o_t \mid q)}{\pi_{\theta_{\mathrm{old}}}(o_t \mid q)}$ is the importance ratio, $\hat{A}_t$ is an advantage estimator, and $r_t(\pi_\theta)$ is the return of executing the policy $\pi_\theta$ at step $t$.

The proof below closely follows Kahn et al. (2017); Ross et al. (2011); Mendonca et al. (2019).

**Definition 1** (Expected Return). *For a fixed task (or a minibatch of tasks) $q$, the total cost of executing $\pi_\theta$ for $H$ steps is the negative return (cost)*

$$J^q(\pi_\theta) := -V_H(\pi_\theta \mid q) = -\mathbb{E}\left[ \sum_{t=1}^H r_t(\pi_\theta \mid q) \right]. \quad (7)$$

Aggregating across tasks gives $J(\pi_\theta) := \mathbb{E}_{q \sim \mathcal{Q}}[J^q(\pi_\theta)] = -\mathbb{E}_{q \sim \mathcal{Q}}[V_H(\pi_\theta \mid q)]$.

**Definition 2** (Hybrid Cost). *For $t \in \{1, \ldots, H\}$ and policies $\pi_1, \pi_2$, define the* hybrid cost

$$J_t^q(\pi_1, \pi_2) := -\mathbb{E}\left[ \sum_{s=1}^t r_s(\pi_1) + \sum_{s=t+1}^H r_s(\pi_2) \right],$$

*so that $J^q(\pi) = J_H^q(\pi, \pi)$ and $J^q(\pi^*) = J_0^q(\pi, \pi^*)$. Intuitively, hybrid cost is the expected cost of executing $\pi_1$ until $t$ and executing $\pi_2$ from $t + 1$ to $H$.*

**Definition 3** (Cost-to-go). *Define*

$$Q_t(\pi_1, \pi_2) := -\mathbb{E}\left[ r_1(\pi_1) + \sum_{s=2}^t r_s(\pi_2) \right],$$

*as a cost to execute $\pi_1$ at initial state and execute $\pi_2$ at the remaining $t - 1$ steps.*

**Lemma C.1** (Policy discrepancy bound). *For $t \in \{1, \ldots, H-1\}$ and policies $\pi_1, \pi_2$,*

$$J_{t+1}^q(\pi_1, \pi_2) - J_t^q(\pi_1, \pi_2) = \mathbb{E}_{o_{\le t} \sim \pi_1}[Q_t(\pi_1, \pi_2) - Q_t(\pi_2, \pi_2)].$$

*Proof.* By definition of $J_t^q$ and conditioning on the random history $o_{<t}$ generated by $\pi_1$, the difference between using $\pi_1$ vs. $\pi_2$ at time $t$ (and $\pi_2$ thereafter) is exactly the displayed quantity. $\square$

**Assumption 1** (Training Error). *There exists $\delta < \infty$ such that for all tasks $q$, steps $t$, histories $o_{<t}$ and outputs $o_t \in \mathcal{O}$,*

$$Q_t(\pi_\theta, \pi_q^*) - Q_t(\pi_q^*, \pi_q^*) \le \delta.$$

Note that for each task $q$, $\pi_q^*$ is an expert actor.

**Assumption 2.** *Let $\hat{\pi}$ be the returned policy from supervised fine-tuning on $\mathcal{D}_{\mathrm{expert}}$ dataset. The supervised training error is bounded by $\epsilon$,*

$$D_{\mathrm{KL}}(\pi_q^* \| \hat{\pi}) \le \sqrt{\varepsilon}.$$

**Theorem C.2** (Bounded suboptimality of the trained policy). *Under Assumptions 1 and 2,*

$$\mathbb{E}_{q \sim \mathcal{Q}}[V_H(\hat{\pi} \mid q)] \ge \mathbb{E}_{q \sim \mathcal{Q}}[V_H(\pi_q^* \mid q)] - \delta H \sqrt{\tfrac{1}{2}\varepsilon}.$$

*Proof.* For any task $q$,

$$J^q(\hat{\pi}) = J_H^q(\hat{\pi}, \pi_q^*) = J_0^q(\hat{\pi}, \pi_q^*) + \sum_{t=0}^{H-1} \left( J_{t+1}^q(\hat{\pi}, \pi_q^*) - J_t^q(\hat{\pi}, \pi_q^*) \right)$$

$$= J^q(\pi_q^*) + \sum_{t=1}^{H} \mathbb{E}_{o_{\leq t} \sim \hat{\pi}} \left[ Q_t(\hat{\pi}, \pi_q^*) - Q_t(\pi_q^*, \pi_q^*) \right] \quad \text{(Lemma C.1)}$$

$$\leq J^q(\pi_q^*) + \delta \sum_{t=1}^{H} \mathbb{E}_{o_{\leq t} \sim \hat{\pi}} \left[ \ell_{o_{\leq t}}(\hat{\pi}, \pi_q^*) \right].$$

From Assumption 2, by Pinsker's inequality and Jensen's inequality we show

$$\mathbb{E}_{q \sim \mathcal{Q}} \mathbb{E}_{o_{\leq t} \sim \hat{\pi}} \left[ \ell_{o_{\leq t}}(\hat{\pi}, \pi_q^*) \right] \leq \mathbb{E}_{q \sim \mathcal{Q}} \mathbb{E}_{o_{\leq t}} \left[ \text{TV}(\hat{\pi}, \pi_q^*) \right] \tag{8}$$

$$\leq \sqrt{\tfrac{1}{2} D_{\text{KL}}\left( \pi_q^* \| \hat{\pi} \right)} \tag{9}$$

$$\leq \sqrt{\tfrac{1}{2}\varepsilon}. \tag{10}$$

Taking $\mathbb{E}_{q \sim \mathcal{Q}}$ yields

$$\mathbb{E}_{q \sim \mathcal{Q}}[J^q(\hat{\pi})] \leq \mathbb{E}_{q \sim \mathcal{Q}}[J^q(\pi_q^*)] + \delta H \sqrt{\tfrac{1}{2}\varepsilon}$$

and substituting cost $J$ back to return $V$ proves the claim. $\qquad \square$

# D    META-PREDICTION DYNAMICS DURING MASA TRAINING

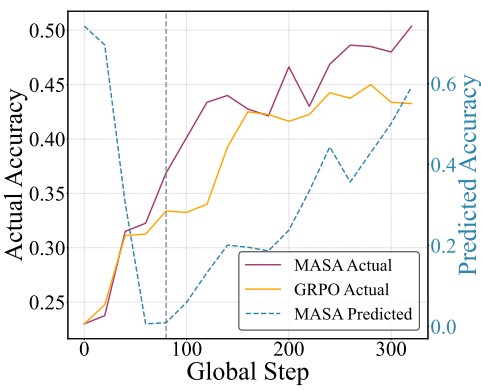 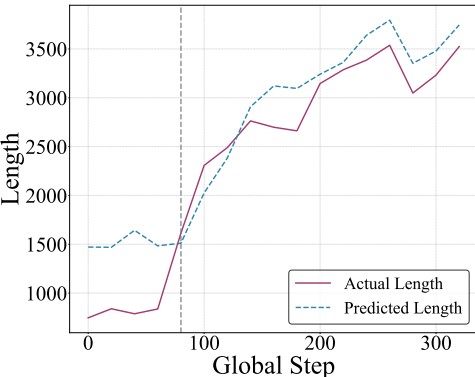

(a) Actual and meta-predicted accuracy over global training steps.

(b) Actual and meta-predicted output length over global training steps.

Figure 7: Actual vs. meta-predicted statistics across training

To analyze the training dynamics of meta-prediction, we tracked how the model's meta-predictions and actual performance changed over the course of training. As shown in fig. 7, the meta-predictions initially differed greatly from the actual values, but the gap gradually narrowed as training progressed.

We also observed an interesting pattern in accuracy meta-prediction. Early in training, the model tended to predict excessively high pass rates for most problems, which created a large discrepancy with the true accuracy, as shown in fig. 7a. This mismatch resulted in low rewards and a sharp drop in the predicted values. Around step 80, the model began to distinguish between easy and hard problems, and MASA's performance improved improved rapidly.

As we can see in fig. 7b, a similar trend appeared in the difficulty metric. At first, the model failed to accurately estimate the token length of correct solutions, but after about step 80 it began to match the actual lengths more closely. Notably, this timing coincided with the point at which MASA began to outperform the baseline, supporting our hypothesis that meta-awareness contributes to performance gains.

# E    USAGE OF LLM

We used LLM to polish writings and to search for related works.

