# OpenReview forum: "Meta-Awareness Enhances Reasoning Models: Self-Alignment Reinforcement Learning"
_ICLR.cc/2026/Conference — ICLR 2026 Conference Withdrawn Submission_

### Official Review · Reviewer_rrks · 2025-10-21

**Soundness:** 2
**Presentation:** 3
**Contribution:** 2
**Rating:** 4
**Confidence:** 4

**Summary:**

The paper proposes MASA (Meta-Awareness via Self-Alignment) - a post-training RL procedure for "reasoning" LLMs in which the model generates, in parallel, (1) task solutions and (2) meta-predictions about its own reasoning (solution length, difficulty/pass rate, and relevant mathematical notions). The meta trajectories are then rewarded for self-consistency with statistics computed from the actual solution rollouts. The authors also introduce MASA-efficient, which adds behavioral cloning of "expert" meta trajectories and active control via predictive gating (filtering near-zero-variance prompts that are either too easy or too hard) and early cutoff (terminating overly long, likely incorrect rollouts). Experiments show improvements on math benchmarks (AIME25, AIME24, AMC23, MATH500, Minerva, Olympiad) with transfer to logic/science/code, as well as faster training compared to GRPO.

**Strengths:**

1. **Clear motivation.** The paper identifies a genuine meta-cognitive gap in existing RL-trained reasoning models and formulates a well-justified hypothesis that self-alignment between predicted and actual reasoning statistics can enhance performance.
2. **Novel use of self-generated meta signals.** MASA’s separation of meta and solution rollouts is creative, enabling meta-cognitive supervision without external annotations or verifiers, which is both scalable and elegant.
3. **Careful Efficiency and Tuning Analysis.** MASA-efficient is rigorously analyzed for efficiency gains, demonstrating real reductions in training time and computation without performance degradation.

**Weaknesses:**

1. **Causality of meta-alignment benefits not conclusively demonstrated.**: The paper shows correlation (Figure 1d) between improved meta-prediction accuracy and reasoning gains, but lacks controlled ablations that isolate which meta components (length, difficulty, notion) actually drive improvement.
2. **Limited model and scale diversity.** All results use Qwen3 models (8B/14B). It remains unclear whether MASA generalizes to other architectures (e.g., Llama-3, Mistral) or smaller models. Demonstrating robustness across architectures and sizes would strengthen the empirical evidence.
3. **Statistical and methodological rigor.** Results are reported as single runs without confidence intervals or standard deviations. Given small datasets such as AIME, variance can be high; statistical significance and multi-seed runs are needed.
4. **Efficiency claims need clearer normalization.** Although MASA-efficient shortens wall-clock time, it doubles rollout generation (meta + solution). Reporting total generated tokens or FLOPs per training step would make comparisons fairer.
5. **Lack of qualitative analysis.** The paper would benefit from concrete examples of failure cases or misaligned meta-predictions to understand when MASA helps or hurts reasoning stability.

**Questions:**

1. **Optimization details:** In Algorithm 1, RL and BC updates use learning rates $\alpha$ and $\beta$. Were these implemented as separate optimizers (e.g., two AdamW instances) or a shared optimizer with scaled gradients?
2. **KL Regularization:** Prior GRPO-based works (e.g., DeepSeek) often include a KL-regularization term for stability over long training runs. Did you experiment with KL control in MASA, and how does varying the KL coefficient affect the stability and accuracy of meta-predictions?
3. **Ablation request:** How does performance change when each meta component (length, difficulty, notion) is removed individually?

---

### Official Review · Reviewer_ebnA · 2025-10-27

**Soundness:** 2
**Presentation:** 2
**Contribution:** 2
**Rating:** 2
**Confidence:** 4

**Summary:**

This paper proposes Meta-Awareness Self-Alignment, where additional sources about the self-awareness of a policy are used as a training signal to improve quality.

**Strengths:**

- The paper is easy to understand, and the proposed method is intuitive.
- Given the experimental setup in the paper, the results are promising.

**Weaknesses:**

- The ablation study is limited. Figure 6 only discusses the share of explained variance when using different reward sources for the meta policy. There are still remaining questions, such as "How would the model perform with the notion awareness (or other components) removed?"
- My main concern is the efficiency of the proposed method, as it requires a significantly larger number of samples. Figure 4 should have addressed this concern, but instead it raises more questions. I don't see a significant improvement from it; it rather seems like noise without any standard deviations. The performance increase at the final point is comparable to the decreases at previous steps. The smoothed plots also make me suspicious about whether the actual plots look even more like noise, and if we are trying to make conclusions from a single training run.
- The training tasks are limited only to math (is this somewhat related to the fact that the notion reward contributed the most to the gains, as evaluated by the authors?). What if my task is not math-related and I could not use the notion reward? Should I use this method then?
- Given the number of proposed hacks in this paper, the lack of source code makes it completely unreproducible.

**Questions:**

See weaknesses

---

### Official Review · Reviewer_EzSr · 2025-10-29

**Soundness:** 2
**Presentation:** 3
**Contribution:** 2
**Rating:** 2
**Confidence:** 4

**Summary:**

The authors propose an approach to encourage the meta-awareness of language models. They propose a method that does two parallel rollouts - one for a meta-path to predict notion, lengths, and difficulty, and the other for predicting solutions similar to typical RL (GRPO). Alternatively, for efficiency, the authros additionally propose an approach to perform (1) predictive gating for zero variance tasks, keeping sampling for only moderately difficult problems and (2) early length cutoff to curtail the lengths of sampled trajectories in RL.

**Strengths:**

- The method is evaluated broadly across Math and out of domain benchmarks.
- Provide some preliminary analysis on the meta-cognitive behaviors found in models trained with GRPO
- Showcase theoretically a bound of the return-to-go of the expert sft procedure they describe.

**Weaknesses:**

- There are several components described but not sufficiently ablated over (e.g what contributes the most for efficiency, the BC objective, the predictive gating, or early length cutoff; also, some reward function definitions)
- The gap between Masa and GRPO are quite small in Table 1 and Table 2(1-2% increase for GRPO with MASA on average for Qwen3-14B), raising the question of the efficacy of the method and whether it is within the bounds of standard error.  It could be useful to run several seeds for each approach to verify that the difference is statistically significant and not due to noise.
- Components of their method have been previously explored (e.g, Zero Advantage should be pruned and explored in DAPO, learning the difficulty of problems is studied in [1], and length-based curricula[2]/mitigations also explored in prior work), making it unclear what the full scope of the contributions of the paper are.
- The authors claim that the model learn "how to think by itself" as stated in the abstract as a differentiating factor from prior work, but this seems a bit of an overclaim with the particular handcrafted rewards that they have designed to train the model in the meta path.
- The efficiency gains as seen in Table 3(a) seem to degrade performance by 3% which is more than the gains that the authors claim for MASA, making it unclear the tradeoff benefit between performance and efficiency.

[1] Learning How Hard to Think: Input-Adaptive Allocation of LM Computation
[2] e3: Learning to Explore Enables Extrapolation of Test-Time Compute for LLMs

**Questions:**

1. How are the notions defined and detected in the method? This is unclear in the writing.
2. How does the behavior of the model qualitatively change with the introduction of the metacognitive behaviors that are described?
3. Address concerns from weaknesses above.

---

### Official Review · Reviewer_rjNp · 2025-10-30

**Soundness:** 2
**Presentation:** 3
**Contribution:** 2
**Rating:** 2
**Confidence:** 3

**Summary:**

The authors propose that aligning the model’s meta-predictions about generated solutions with their actual characteristics can improve performance. They target length, difficulty, and the list of math notions used, predicted via a separate prompt. These predictions are then used to build an efficient algorithm that discards problems deemed unhelpful. The authors report performance gains across various setups, along with improved sampling efficiency.

**Strengths:**

* I like the idea of using such aux signals to improve performance.
* To the best of my knowledge, the idea is novel and interesting.

**Weaknesses:**

While I acknowledge the strengths, I have serious concerns about the methodology:

* **Statistical significance**
  Gains of a few absolute points on math benchmarks are often within noise. Please add standard deviations to all tables. In particular, Table 2 currently suggests at most that MASA does not change performance on OOD tasks. Also, Figure 4 indicates the runs have not reached a performance plateau -- the appropriate point for comparison. Please report results at convergence and include confidence intervals over training time.

* **Other models**
  To support generality, please include experiments with LLaMA models.

* **Efficiency comparison**
  Could MASA require an additional 18 hours of training to close the AIME gap (Table 3)? Please extend Figure 4(c) to 50 hours so it aligns with Table 3 and clarifies the time–performance trade-off.

**Questions:**

*  In authors' experiments, GRPO often outperforms DAPO, which conflicts with the DAPO paper’s findings. Please explain this discrepancy.

---

### Note · Authors · 2026-01-04

**Comment:**

We appreciate the reviewers' effort for providing constructive feedback. As we are planning to polish our paper based on the feedbacks, we withdraw from ICLR 2026 submission.

**Withdrawal Confirmation:**

I have read and agree with the venue's withdrawal policy on behalf of myself and my co-authors.